# Synergistic effects of surfactant blends on lignite dust wettability

**Xinhui Luo[1], Xueming Fang[1,2]\*, Jie Liu[1,2,3], Henglin Liu[1,3], Jiajia Zou[1], Yuanxi Xu[1], Xingxing Liang[1], Wei Sun[1]**

**1** Faculty of Public Security and Emergency Management, Kunming University of Science and Technology, Kunming, Yunnan, China, **2** Yunnan Institute of Public Safety and Emergency Management, Kunming, Yunnan, China, **3** Key Laboratory of Geological Disaster Risk Prevention and Emergency Reduction, Kunming, Yunnan, China

\* fxm08045304@163.com

## Abstract

In this study, we employed a combination of theoretical and experimental analyses to explore the effects of the physico-chemical properties of lignite samples and surfactants on lignite dust's wettability, thereby improving dust control in coal mines. First, we measured and analysed the coal samples' industrial composition, elemental composition and chemical structure. It was found that the selected lignite dust has high ash and low moisture content and contains many hydrophobic functional groups, resulting in poor wettability by water. Next, we conducted surface tension tests, contact angle tests and lignite dust settling experiments to screen 12 monomer surfactants, exploring the impact of these solutions on lignite dust wettability. Finally, considering all monomer surfactants' abilities to reduce surface tension, decrease contact angles and promote dust settling in solutions, we selected five surfactants (AES, MES-30, AEO-9, CDEA and CHSB) for blending based on their excellent performance in tests. We prepared the blends of these five surfactants, each with a mass fraction of 0.5 wt%, in a 1:1 ratio, resulting in 10 blended solutions. We measured the performance of these solutions and revealed that the AES+AEO-9 blend demonstrated a significant synergistic effect, markedly enhancing the capture efficiency of water for lignite dust.

## 1 Introduction

As of 2024, China has discovered 173 types of mineral resources, with substantial total reserves [1], playing an irreplaceable role in the country's social development and economic growth. In 2023, China's total primary energy production of standard coal reached 4.83 billion tons, a 4.2% increase from the previous year, of which coal account for 66.6% [2]. Coal mining releases a significant amount of coal dust, polluting the working environment, and excessive inhalation by miners can cause

**Data availability statement:** All relevant data are within the paper and its Supporting Information files.

**Funding:** This research was financially supported by the Teacher category project of Yunnan Provincial Department of Education Scientific Research Fund (grant No. 2023J0158), Yunnan Fundamental Research Projects (grant NO. 202401AU070195), Scientific Research Fund Projects of Yunnan Provincial Department of Education (grant NO. 2024Y133), Teacher category project of Yunnan Provincial Department of Education Scientific Research Fund (grant No. 2023J0157), Key Research and Development Plan of Yunnan Province (grant No. 202303AA080014), College Students' Innovative Entrepreneurial Training Plan Program (grant No. 202310674017). These funders provide financial support for experimental research and play a role in the decision to publish the manuscript.

**Competing interests:** The authors have declared that no competing interests exist.

irreversible pneumoconiosis [3,4]. Additionally, coal dust suspended in the air can easily explode, posing a considerable safety hazard and necessitating effective measures to reduce coal dust hazards.

Dust control techniques are relatively advanced, with common methods including ventilation dust removal, spray dust suppression, dust collectors, coal seam water injection, foam dust suppression, air curtain dust barriers and chemical dust suppression [5]. Coal mines typically use spray water and mine ventilation to suppress dust dispersion, but the hydrophobic nature of coal dust makes pure water inefficient for dust suppression [6–8]. Surfactants, known for their excellent wetting and solubilising properties, play a crucial role in dust control, firefighting and oil extraction [9]. In coal dust suppression, surfactants are often reducing the surface tension of water and enhancing the wettability of coal dust [10]. Wang et al. [11] demonstrated that lignite dust samples' wettability was most related to pore size, with smaller average pore sizes leading to poorer wettability, the anionic surfactant AES significantly improved the wettability of lignite. Hu et al. [12] used both numerical and experimental analyses to confirm that carbonyl, ether, and carboxyl groups in surfactants are key factors influencing the wettability of lignite. To construct macromolecular model of coal and uncover the mechanism of action of compounded surfactants on the wettability of coal dust, Jing et al. [13] used molecular simulation, theoretical analysis and experimental analysis. Jiang et al. [14] used theoretical analysis and molecular simulation to reveal the order of the wetting ability of water molecules for the functional groups of coal. Tang and Zhao [15] found that non-ionic surfactants enhanced wettability by increasing the adsorption capacity on the hydrophobic shell of dust particles. Han and Wang et al. [16,17] found that the addition of electrolytes to a low concentration surfactant solution significantly reduced the surface tension and enhanced the ability of the composite surfactant to improve coal wettability.. Hao et al. [18] demonstrated that the nonionic-anionic complex system exhibited the most significant longitudinal distribution between coal and water, forming a denser adsorption layer and achieving a superior wetting effect compared to monomer surfactants. Wang et al. [19] found that short-term soaking of coal dust in water increased hydroxyl content, enhancing its wetting properties. This enhancement also exhibited a synergistic effect when combined with surfactant action. Han et al. [20] used wetting area measurements to characterize droplet behavior on coal dust, confirming the excellent wetting performance of SDBS.

Although extensive research has been conducted on the synergistic effects of surfactant combinations on coal dust wettability, practical engineering applications remain limited. Further in-depth research is needed on the physico-chemical properties of coal dust and the impact and mechanism of the surfactant blend on the coal dust wettability. In this study, we focused on lignite from the Xiaolongtan coal mine, where we selected 12 surfactants to investigate the effects of surfactants and their blends on coal dust wettability, as well as the changes in the adsorption state of surfactant molecules during contact angle, surface tension and settling experiments. The results can provide some reference for selecting and designing surfactants in coal dust wettability.

## 2 Materials and methods

### 2.1 Experimental material

We sourced the coal samples used in the experiments from Xiaolongtan Town, Kaiyuan City, Yunnan Province. We crushed and ground the samples using a mill to pass through an 80-mesh (0.178 mm) sieve for subsequent experiments. In this study, we selected 12 surfactants based on non-toxicity, non-irritation, non-flammability principles and economic and environmental friendliness [21]. For the experiments, we diluted these surfactants into solutions with seven different mass fractions: 0.01 wt%, 0.03 wt%, 0.05 wt%, 0.07 wt%, 0.1 wt%, 0.3 wt% and 0.5 wt%.

Because cationic surfactants typically require high concentrations to form easily wettable adsorption layers [4], they are not suitable as wetting agents for dust in this context. Therefore, we selected no cationic surfactants for this experiment. The surfactants used in the experiment included four anionic, four non-ionic and four amphoteric surfactants, as shown in **Table 1**.

### 2.2 Experimental methods and equipment

We measured the lignite dust's composition using a WGLL-625BE oven (Taisite (Tianjin) instruments Co., Ltd., Tianjin, China, as shown in **Fig 1a**) and a XL-2000 Box type high temperature furnace (Tianjian (Hebi) Technology Co., Ltd., Henan, China, as shown in **Fig 1b**). We analysed the lignite using ONH836 (as shown in **Fig 1c**) and CS844 (as shown in **Fig 1d**) elemental analysers manufactured by Leco Corporation (St. Joseph, MI, USA). We analysed the surface functional groups of the coal samples using an ALPHA infrared spectrometer (Bruker Corporation, Mannheim, Germany, as shown in **Fig 1e**). For infrared spectroscopy, we mixed 2 mg of lignite dust with 200 mg of KBr in an agate mortar and ground to below 2 μm. We used the KBr tablet method for infrared spectral detection of the lignite dust.

We measured the surface tension of various solutions using an MC-1021 liquid surface tensiometer (Mincee Instrument Co., Ltd., Xiamen, China, as shown in **Fig 1f**) with the ring method, following the GB6541–86 standard. We pressed the ground coal dust into thin tablets using an HY-12 tablet press mould (Tianjin Tianguang Optical Instrument Co., Ltd., Tianjin, China) at a pressure of 20 MPa for 2 min. These thin tablets were used for contact angle measurement experiments. We measured the contact angles between the surfactant solutions or distilled water and the coal sample tablets using a ZJ-6900 optical water droplet contact angle metre (Zhijia Instrument Co., Ltd., Shenzhen, China, as shown in **Fig 1e**) with the sessile drop method. We measured the contact angle and surface tension at room temperature (25°C).

**Table 1. The 12 kinds of surfactants selected in the experiment.**

| Ionic Type | Name | Chemical Formula | Abbreviation |
|---|---|---|---|
| Anionic | Sodium Dodecyl Sulfate | $C_{12}H_{25}SO_4Na$ | SDS |
| Anionic | α-Olefin Sulfonate | $C_{17}H_{34}ONaS$ | AOS |
| Anionic | Fatty Alcohol Polyoxyethylene Ether Sulfate | $C_{14}H_{33}NO_5NaS$ | AES |
| Anionic | Sodium Lauryl Polyoxyethylene Ether Sulfosuccinate | $C_{22}H_{40}O_{10}Na_2S$ | MES-30 |
| Non-ionic | Hexyl D-Glucoside | $C_{17}H_{34}O_3NaS$ | APG-06 |
| Non-ionic | Polyoxyethylene Sorbitan Monooleate | $C_{24}H_{44}O_6$ | Tween-80 |
| Non-ionic | Fatty Alcohol Polyoxyethylene Ether 9 | $(C_2H_4O)nC_{12}H_{24}O$ | AEO-9 |
| Non-ionic | Coconut Oil Fatty Acid Diethanolamide | $C_{16}H_{33}NO_3$ | CDEA |
| Amphoteric | Cocoamidopropyl Betaine | $C_{20}H_{36}N_2O_6Na_2$ | CAB-35 |
| Amphoteric | Sodium Lauryl Glutamate | $C_{17}H_{30}NO_5Na$ | LG-95P |
| Amphoteric | Cocoamidopropyl Hydroxysultaine | $C_{20}H_{42}N_2O_5S$ | CHSB |
| Amphoteric | Lauramidopropyl Betaine | $C_{19}H_{40}N_2O_2$ | LAB-35 |

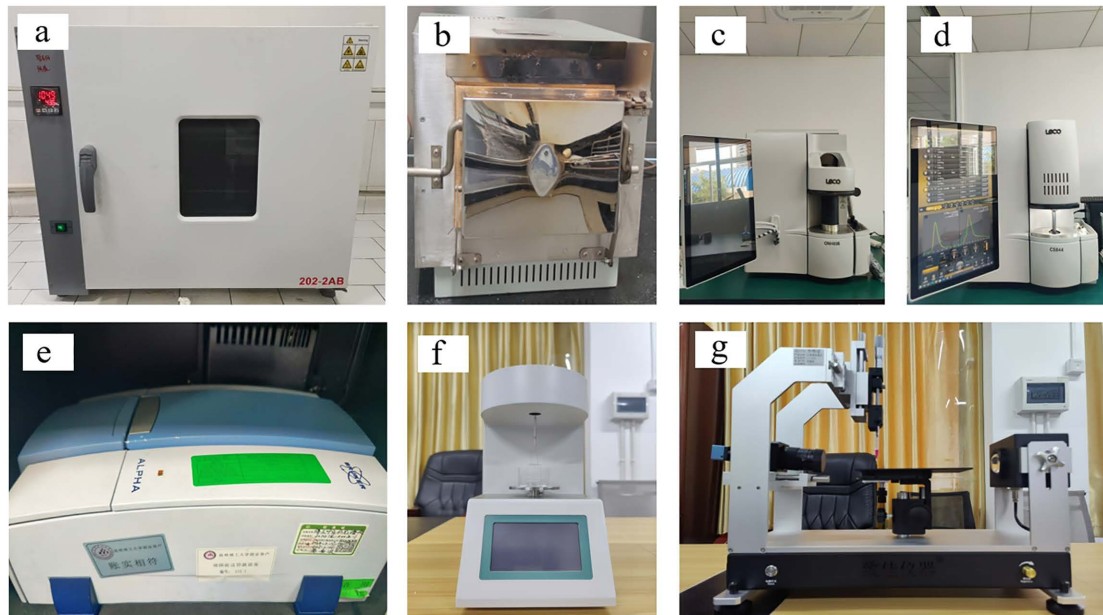

**Fig 1. Diagram of the experimental equipment.**

According to the 'Performance Determination Method for Dust Suppressants Used in Mines (MT506-1996)' [22], we assembled a coal dust settling experiment apparatus using an iron stand, weighing bottles, metal rings, fast qualitative filter paper, glass funnels and positioning rings. We used the natural settling method to measure lignite dust's settling speed in different solutions (as shown in **Fig 2**), reflecting the wetting ability of different solutions on the coal dust. In the experiment, 10 mg of coal dust was allowed to fall naturally through a glass funnel and form a conical pile on fast qualitative filter paper. The positioning ring was moved, and the hook arm was rotated to make the filter paper contact the solution. The timing started when the filter paper contacted the solution and stopped when all the lignite dust had settled [11]. Each settling test, surface tension measurement and contact angle test was repeated three times, and the average value was taken. (The results of each settling test needed to be within ±7% of the average value to be considered valid.)

## 3 Results and discussion

### 3.1 Influence of physico-chemical properties of coal dust on lignite dust wettability

We conducted the elemental and industrial analyses of lignite samples on an air-dried basis. The industrial analysis measured four indicators in lignite dust: moisture, ash, volatile matter and fixed carbon. The elemental analysis determined the mass percentages of five primary elements (carbon, hydrogen, oxygen, nitrogen and sulfur) in the coal samples. **Fig 3** shows the results.

The total proportion of free and bound water in the sample was 23.76%, with free water constituting a larger share [23]. The ash content in the coal dust was 17.24%, and the volatile matter content was 39.04%. The sample dust exhibited a hydrophobic state because dust wettability is positively correlated with its ash content and negatively correlated with its volatile matter content [24,25]. The fixed carbon content in the sample was 19.96%, indicating a low degree of coalification for the lignite sample.

The carbon element had the highest mass fraction in the sample coal dust, accounting for 36.93%, which is one of the main reasons for the poor wettability of lignite coal dust. The oxygen element content was 25.4%, imparting some

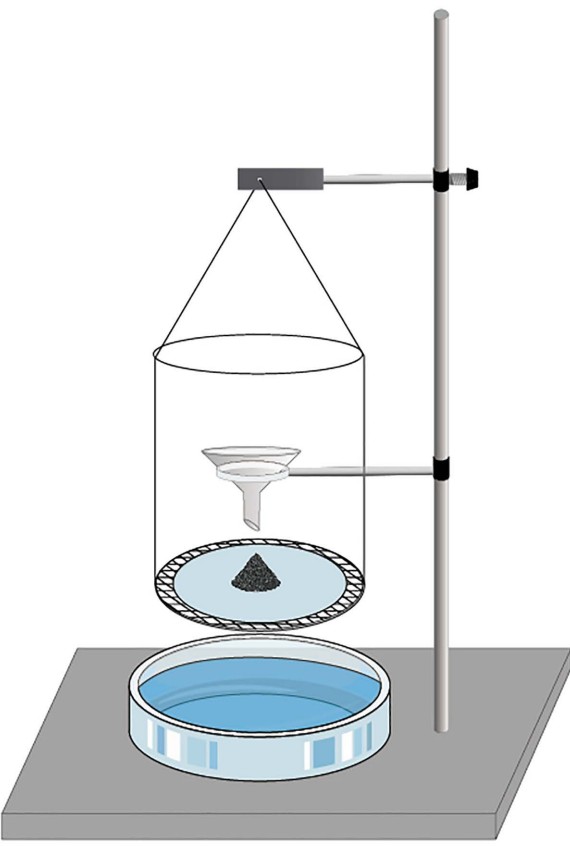

**Fig 2. Sedimentation experimental device.**

hydrophilicity to the lignite coal dust. The hydrogen element content was 1.59%. Hydrogen is typically present in the side chains and functional groups of coal molecules. It can help improve coal dust's wettability. However, the relatively low hydrogen content also contributes to the poor wettability of lignite coal dust.

Lignite dust's complex chemical structure is a key factor affecting its wettability. We conducted Fourier transform infrared (FTIR) spectroscopy experiments on lignite dust and analysed the chemical structure of the lignite dust by referring to the absorption peak assignment table for relevant groups (as shown in **Table 2**). **Fig 4** plots the results based on FTIR experiments. The peak positions for lignite are relatively simple, primarily including aliphatic hydrocarbons, aromatic hydrocarbons and oxygen-containing functional groups. The content of carbonyl and aromatic hydrocarbons is relatively high, whereas that of aliphatic hydrocarbons is low. A large number of hydrophobic groups, such as aromatic hydrocarbons and aliphatic hydrocarbons, endow lignite with strong hydrophobic characteristics, whereas the hydroxyl and oxygen-containing functional groups present in lignite impart some degree of hydrophilicity.

### 3.2 Effect of monomer surfactants on coal dust wettability

**3.2.1 Surface tension of monomer surfactants.** The surface tension of distilled water was measured to be 71.60 mN/m at room temperature. **Fig 5** shows the surface tension data for the solutions of 12 monomer surfactants.

Moreover, **Fig 5** shows that all 12 monomer surfactants effectively reduce the surface tension of their solutions because surfactant molecules contain both hydrophobic groups composed of non-polar hydrocarbon chains and hydrophilic groups

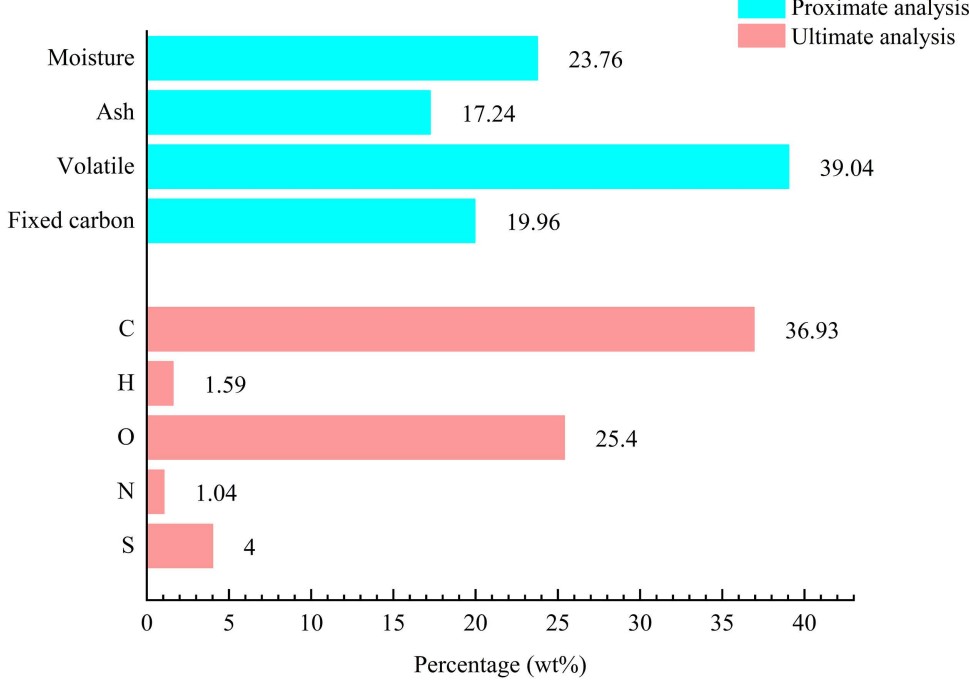

**Fig 3. Ultimate analysis and proximate analysis of lignite.**

**Table 2. Absorption peaks of groups.**

| Wave Number (cm⁻¹) | Corresponding Group Type |
|---|---|
| 3750–3000 | Hydroxyl |
| 2920–2860 | Aliphatic Hydrocarbons |
| 1900–1650 | Carbonyl |
| 1590–1470 | Most Aromatic Hydrocarbons |
| 900–690 | Monosubstituted Aromatic Hydrocarbons |

composed of polar groups. In aqueous solutions, the hydrophilic and hydrophobic groups orient towards the water and air, respectively, thus reducing the solution's surface tension. MES-30, CDEA and LG-95P are the most effective surfactants at lowering the surface tension, each reducing it to below 30 mN/m. Surfactants such as SDS, AOS, AES, APG-06, AEO-9, CAB-35, CHSB and LAB-35 have a lesser impact on surface tension, reducing it from 71.60 mN/m to between 30 and 40 mN/m.

As the solution concentration increases, the surface tension change for all solutions tends to plateau because excess surfactant molecules aggregate to form micelles, and the surface tension hardly decreases further. At this point, the surfactant concentration reaches the critical micelle concentration (CMC). **Fig 5** shows that the CMC for the four anionic surfactants (SDS, AOS, AES and MES-30) and two non-ionic surfactants (APG-06 and Tween-80) is 0.1 wt%. The four amphoteric surfactants have a lower CMC of 0.07 wt%, whereas the non-ionic surfactants AEO-9 and CDEA have the lowest CMC at 0.01 wt%.

The performance of monomer surfactants can be characterised by two indicators: the CMC and the ability to reduce surface tension. Non-ionic surfactants AEO-9 and CDEA can quickly reach their CMC and significantly reduce the solution's surface tension at CMC, with 54.48% and 55.94% reduction rates, respectively.

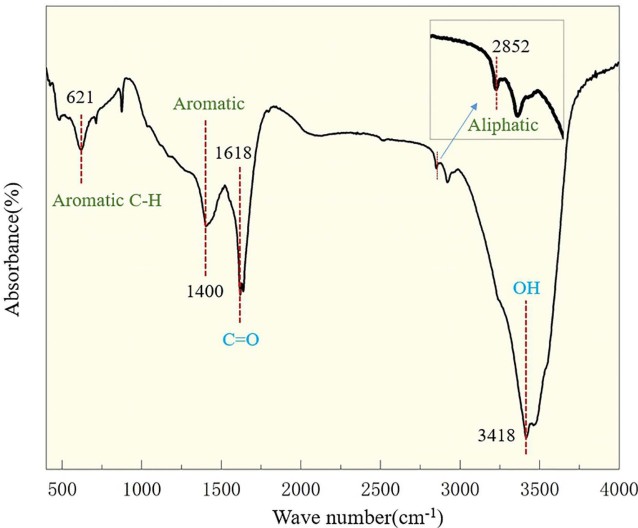

**Fig 4. Infrared spectrogram of lignite.**

**3.2.2 Contact angle of monomer surfactants with lignite.** The contact angle between dust tablets and solutions intuitively represents the solutions' wettability on the dust. When a droplet spreads and stabilises on a solid interface, the relationship between the interfacial tensions of solid–liquid, solid–gas and gas–liquid, and the contact angle θ is shown in **Fig 6**. The smaller the contact angle between the liquid and the solid, the stronger the solution's wettability on the solid. A contact angle＞90° indicates that the solution cannot wet the solid, whereas a contact angle＜90° indicates that the solution can wet the solid. **Fig 7** shows the contact angles between the solutions of 12 surfactants and the lignite tablets.

At room temperature (25°C), the contact angle between distilled water and the lignite tablet was measured to be 87.81°, confirming the poor lignite dust wettability. As the concentration of the solutions increases, the contact angle between the solution and the lignite tablets gradually decreases.

When the mass fraction of AES, AOS and CDEA solutions reached 0.01 wt%, the contact angle between the solutions and the coal dust tablets dropped below 40°, indicating that these three surfactants have an excellent ability to improve dust wettability even at low concentrations, with AES being the most prominent, reducing the contact angle by 62.64%. When the mass fraction of the four amphoteric surfactants (CAB-35, LG-95P, CHSB and LAB-35) reached 0.5 wt%, their solutions had contact angles with coal dust tablets in the 20°–30° range.

At the same concentration, AOS, AES, MES-30, AEO-9 and CDEA showed more significant improvements in dust wettability compared to the amphoteric surfactants, with contact angles between 10° and 20° for solutions and coal dust tablets. Among them, AOS solution had the smallest contact angle, with the 0.5 wt% AOS solution having a contact angle of only 10.21° with the lignite tablet, representing an 88.37% reduction compared with the contact angle between distilled water and the lignite tablet. However, surfactants SDS, APG-06 and Tween-80 had a relatively poor ability to enhance the lignite dust's wettability when their concentrations reach CMC, with contact angle reduction rates in the 53%–65% range.

**3.2.3 Sedimentation experiment of lignite dust in monomer surfactants.** We measured the sedimentation speed of lignite dust in monomer surfactants using a self-assembled dust sedimentation experimental apparatus. We used a 10-mg sample of lignite dust for the sedimentation experiment. If complete sedimentation did not occur within 90 min, the lignite dust was considered non-settling in the solution (Use/ to indicate that the lignite dust does not settle in the solution). Table 3 shows the experimental results.

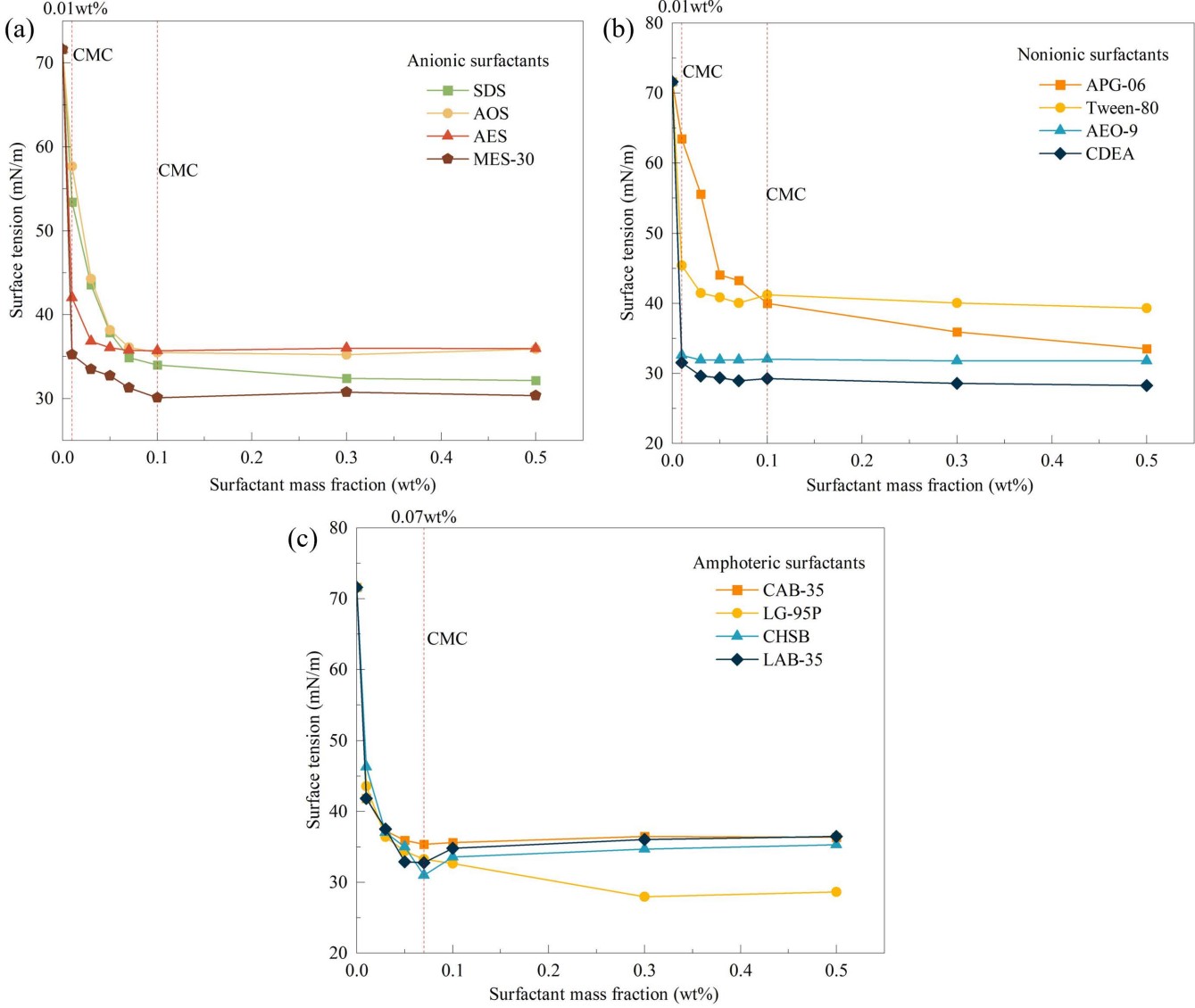

**Fig 5. Surface tension of surfactants at different concentrations;(a) Anionic surfactants, (b) Nonionic surfactants, (c) Amphoteric surfactants.**

**Table 3** shows that the lignite dust's sedimentation speed in most surfactant solutions increased with increased solute concentration. Lignite dust did not settle in SDS, AOS, MES-30, APG-06, Tween-80, CDEA, CAB-35, LG-95P, CHSB and LAB-35 solutions with a mass fraction of 0.01 wt%, but it settled smoothly in AES and AEO-9 solutions. The surfactant solutions of APG-06, LG-95P and LAB-35 showed poor wetting effects on lignite dust at concentrations not exceeding 0.1 wt%. Additionally, when the concentrations of these three solutions increased to 0.5 wt%, the sedimentation speed of lignite dust in the solutions did not significantly improve, with speeds of $2.35 \times 10^{-3}$ mg/s, $1.95 \times 10^{-3}$ mg/s and $2.959 \times 10^{-3}$ mg/s, respectively.

AES and AEO-9 solutions demonstrated a strong ability to improve the wetting characteristics of lignite dust, enabling rapid sedimentation of lignite dust in solutions with seven different mass fractions. However, lignite dust did not completely

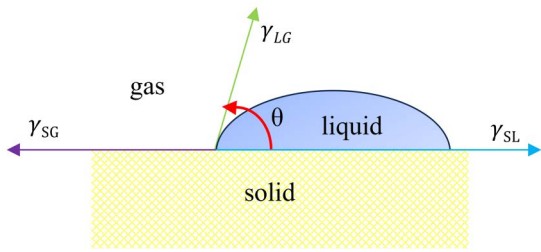

**Fig 6. Contact angle between solid and liquid.**

**Fig 7. Contact angle of surfactants at different concentrations;(a) Anionic surfactants, (b) Nonionic surfactants, (c) Amphoteric surfactants.**

Table 3. Sedimentation velocity of surfactants at different concentrations.

| Category | Sedimentation velocity (× 10⁻³ mg/s) | | | | | | |
|---|---|---|---|---|---|---|---|
| | 0.01 wt% | 0.03 wt% | 0.05 wt% | 0.07 wt% | 0.1 wt% | 0.3 wt% | 0.5 wt% |
| SDS | / | / | / | 2.014 | 2.516 | 3.609 | 5.689 |
| AOS | / | 2.430 | 2.724 | 3.962 | 4.749 | 7.063 | 37.024 |
| AES | 10.327 | 13.311 | 15.219 | 17.514 | 26.632 | 33.316 | 37.024 |
| MES-30 | / | / | / | 2.051 | 5.069 | 10.080 | 11.105 |
| APG-06 | / | / | / | / | / | 1.973 | 2.347 |
| Tween-80 | / | / | / | / | / | / | / |
| AEO-9 | 52.398 | 54.132 | 60.697 | 66.647 | 74.013 | 76.927 | 90.918 |
| CDEA | / | 2.4607 | 3.0171 | 4.0147 | 4.6138 | 5.5918 | 6.6287 |
| CAB-35 | / | / | / | 1.929 | 2.326 | 2.711 | 2.851 |
| LG-95P | / | / | / | / | / | 1.853 | 1.947 |
| CHSB | / | / | 1.865 | 2.151 | 4.732 | 5.079 | 5.143 |
| LAB-35 | / | / | / | / | / | 2.762 | 2.959 |

settle within 90 min in any of the seven different mass fractions of Tween-80 solutions, indicating that Tween-80 is unsuitable for promoting the sedimentation of lignite dust in solutions and should not be selected for blending.

The sedimentation experiments revealed that AEO-9 had the strongest ability to promote the sedimentation of lignite dust among the 12 surfactants used. The sedimentation speed of lignite dust in a 0.5 wt% AEO-9 solution was $90.92 \times 10^{-3}$ mg/s. The sedimentation performance of lignite dust in anionic surfactant solutions was better than that in non-ionic and amphoteric surfactant solutions.

### 3.3 Effect of compound surfactants on coal dust wettability

The aforementioned experiments indicated that AES, MES-30, AEO-9, CDEA and CHSB among the selected 12 surfactants significantly enhance the coal dust wettability. They quickly reach the CMC, reduce surface tension, decrease contact angle and promote rapid sedimentation of coal dust in solutions. Thus, these five surfactants, with a mass fraction of 0.5 wt%, were chosen for 1:1 blending. Surface tension, contact angle and dust sedimentation experiments were conducted with the compound solutions, and **Fig 8** shows the data.

After blending, most compound solutions exhibited a synergistic effect on surface tension, although the changes were minor. The surface tension of all compound solutions was below 35 mN/m. The surface tensions of AES＋CDEA and AES＋AEO-9 compound solutions were 26.58 mN/m and 27.47 mN/m, respectively, with reduction rates higher than 60% compared with distilled water, making them the two compound solutions with the lowest surface tension. The surface tension of CDEA blended with the other four surfactants decreased by 1–2 mN/m. The AES＋MES-30 combination, being anionic＋anionic, had a surface tension of 31.17 mN/m between the values of AES and MES-30 alone, showing no synergistic effect. The surface tension of AES＋AEO-9 was 34.65 mN/m, the weakest in reducing solution surface tension.

The changes in contact angles after blending were minor because the contact angles between the 0.5 wt% single surfactant solutions and lignite tablets were already low. The contact angles of MES-30＋AEO-9, AES＋CHSB and MES-30＋CHSB compound solutions with coal dust tablets were between those of their components, being 13.89°, 20.30° and 18.69°, respectively. AES＋MES-30, AEO-9＋CDEA and AES＋CDEA compound solutions performed poorly in contact angle tests, particularly AES＋CDEA, which increased the contact angle by 8.16° compared with the better-performing single component, reducing the solution's wettability on lignite dust. AES＋AEO-9, MES-30＋CDEA and CDEA＋CHSB compound solutions showed synergistic effects, with contact angles decreasing to below 10°, specifically 9.99° and 9.97° for AES＋AEO-9 and CDEA＋CHSB, respectively.

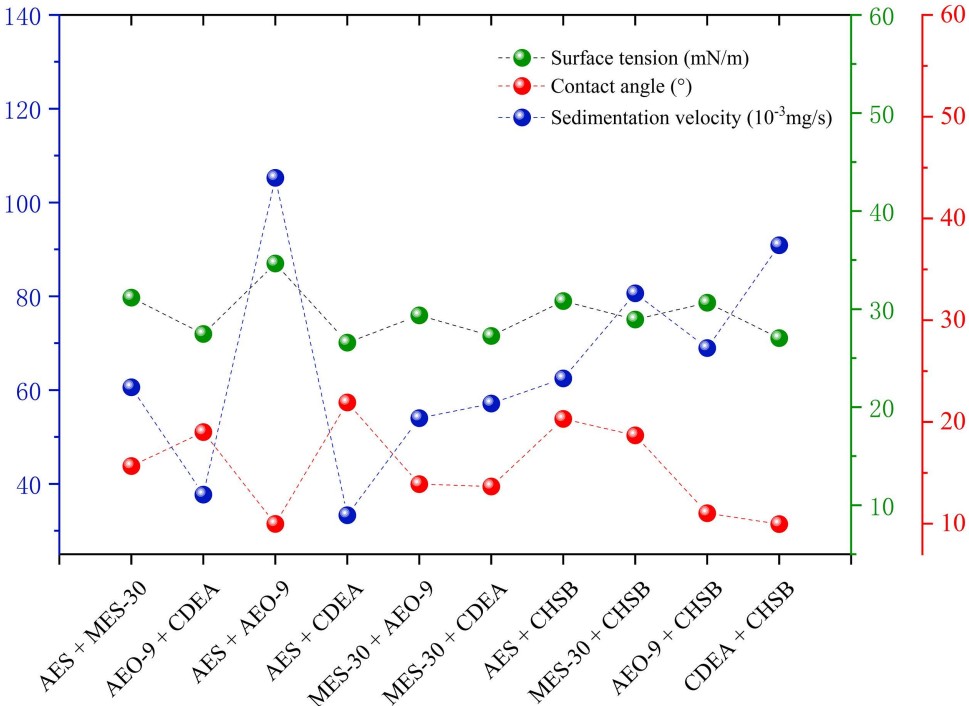

**Fig 8. The wetting feature of compound solutions.**

Compared with single surfactant solutions, the sedimentation speed of lignite dust in compound solutions significantly improved. The sedimentation speeds in AES + MES-30, AES + AEO-9, MES-30 + CDEA, AES + CHSB, MES-30 + CHSB and CDEA + CHSB solutions increased, with changes ranging from 15 to 85 mg/s. The sedimentation speed increased by approximately $84 \times 10^{-3}$ mg/s, 14 times higher, for the CDEA + CHSB blend compared to the CDEA single surfactant solution. Despite the CDEA + CHSB combination showing the best improvement, the fastest sedimentation speed was observed in the AES + AEO-9 compound solution at $105.26 \times 10^{-3}$ mg/s.

## 4 Conclusion

1. The Xiaolongtan lignite dust's wettability is significantly affected by its composition, elemental content and chemical structure. The high ash content, low moisture content and numerous hydrophobic functional groups in the coal dust result in poor wettability, making simple water spray dust suppression inefficient for coal mine dust control.

2. Experiments revealed that increasing the mass fraction of surfactants decreases the solution's surface tension and the contact angle between the solution and dust tablets while increasing the sedimentation speed of lignite dust in the solution. AES, MES-30, AEO-9, CDEA and CHSB have a remarkable ability to enhance coal dust wettability and were selected as components for compound solutions.

3. Performance measurements of compound surfactant solutions showed that the AES + AEO-9 blend exhibits a good synergistic effect, with a surface tension of 34.65 mN/m, a contact angle of 9.99°, and a settling rate of coal dust in the solution of $105.26 \times 10^{-3}$ mg/s, making it suitable as a dust suppression additive for Xiaolongtan lignite dust, significantly enhancing the capture efficiency of lignite dust.

## Supporting information

**S1 Data.** Experimental data.
(ZIP)

## Author contributions

**Conceptualization:** Xinhui Luo, Xueming Fang.

**Data curation:** Xinhui Luo, Henglin Liu.

**Formal analysis:** Xinhui Luo, Jie Liu.

**Funding acquisition:** Xinhui Luo, Xueming Fang.

**Investigation:** Xinhui Luo, Jiajia Zou, Yuanxi Xu.

**Methodology:** Xueming Fang, Jie Liu.

**Validation:** Xinhui Luo, Yuanxi Xu.

**Visualization:** Xinhui Luo, Xueming Fang, Jiajia Zou.

**Writing – original draft:** Xinhui Luo, Xueming Fang.

**Writing – review & editing:** Jie Liu, Henglin Liu, Xingxing Liang, Wei Sun.

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
