## [Decision Letter · Decision Letter 0]

21 May 2025

PONE-D-25-21428Synergistic effects of surfactant blends on lignite dust wettabilityPLOS ONE

Dear Dr. Fang,

Thank you for submitting your manuscript to PLOS ONE. After careful consideration, we feel that it has merit but does not fully meet PLOS ONE’s publication criteria as it currently stands. Therefore, we invite you to submit a revised version of the manuscript that addresses the points raised during the review process.

This manuscript has been reviewed. The reviewer's comments are included at the bottom of this letter.

The reviewer(s) would like to see moderate revisions made to your manuscript before publication.

We look forward to receiving your revised manuscript.

Kind regards,

Ugur Ulusoy, Ph.D.

Academic Editor

PLOS ONE

“This research was financially supported by the Teacher category project of Yunnan Provincial Department of Education Scientific Research Fund (grant No. 2023J0158), Yunnan Fundamental Research Projects (grant NO. 202401AU070195), Scientific Research Fund Projects of Yunnan Provincial Department of Education (grant NO. 2024Y133), Teacher category project of Yunnan Provincial Department of Education Scientific Research Fund (grant No. 2023J0157)�Key Research and Development Plan of Yunnan Province�grant No. 202303AA080014�, College Students' Innovative Entrepreneurial Training Plan Program�grant No. 202310674017�.”

Reviewers' comments:

Reviewer's Responses to Questions

**Comments to the Author**

1. Is the manuscript technically sound, and do the data support the conclusions?

Reviewer #1: Yes

Reviewer #2: Yes

Reviewer #3: Yes

Reviewer #4: Yes

2. Has the statistical analysis been performed appropriately and rigorously? 

Reviewer #1: Yes

Reviewer #2: Yes

Reviewer #3: Yes

Reviewer #4: Yes

3. Have the authors made all data underlying the findings in their manuscript fully available?

Reviewer #1: Yes

Reviewer #2: Yes

Reviewer #3: Yes

Reviewer #4: Yes

4. Is the manuscript presented in an intelligible fashion and written in standard English?

Reviewer #1: Yes

Reviewer #2: Yes

Reviewer #3: Yes

Reviewer #4: Yes

5. Review Comments to the Author

Reviewer #1: This study proposes a set of solutions to the coal dust pollution problem in Xiaolongtan Town. Specifically, it first analyzes the physical and chemical properties of the coal dust, and then optimally selects the dust suppressants suitable for the local coal dust through experiments. However, there are some issues with the language description in the article, which may cause confusion.

Detailed specific comments

1.In the text: 1 Introduction, first paragraph. In coal dust suppression. Surfactants are often wetting agents because they reduce surface tension and enhance wettability [10].

Comment: The main content of this study is to solve the problem of coal dust pollution. Therefore, the description here can be written as "reducing the surface tension of water and enhancing the wettability of coal dust."

2.In the text: 2.2 Experimental methods and equipment, second paragraph. We pressed the ground coal dust into thin tablets using an HY-12 tablet press mould (Tianjin Tianguang Optical Instrument Co., Ltd., Tianjin, China) at a pressure of 20 MPa for 2 min.

Comment: When pressing tablets for contact angle measurement, the specific dimensions of the pressed coal tablets should be stated.

3.In the text: 2.2 Experimental methods and equipment, second paragraph. We measured the contact angles between the surfactant solutions or distilled water and the coal sample tablets using a ZJ-6900 optical water droplet contact angle metre (Zhijia Instrument Co., Ltd., Shenzhen, China, as shown in Fig. 1e) with the sessile drop method.

Comment: When measuring the contact angle using the sessile drop method, the choice of which second of data should be expressed.

4.In the text: 3.1 Influence of physico-chemical properties of coal dust on lignite dust wettability, second paragraph. The total proportion of free and bound water in the sample was 23.76%, with free water constituting a larger share.

Comment: In your figures, there is no visible indication that free water accounts for a relatively large proportion. If this analysis is based on past experience, relevant references should be added.

5.In the text: 3.2.1 Surface tension of monomer surfactants, fourth paragraph. Non-ionic surfactants AEO-9 and CDEA can quickly reach their CMC and significantly reduce the solution’s surface tension at CMC concentration, with 54.48% and 55.94% reduction rates, respectively.

Comment: CMC already represents the critical micelle concentration, so "CMC concentration" is a redundant expression.

6.In the text: 3.2.2 Contact angle of monomer surfactants with lignite, second paragraph. As the concentration of the solutions increases, the contact angle between the solution and the lignite tablets gradually decreases. At room temperature (25°C), the contact angle between distilled water and the lignite tablet was measured to be 87.81°, confirming the poor lignite dust wettability.

Comment: These two sentences lack logical coherence. The first part describes that the contact angle decreases as the solution concentration increases, while the second part mentions that the contact angle of water at room temperature is quite large. There is no connection with the description in the third paragraph about surfactants reducing the contact angle. It would be better to swap the order of these two sentences and combine them with the content in the third paragraph for a more coherent narrative.

7.In the text: 3.2.2 Contact angle of monomer surfactants with lignite, fourth paragraph. However, surfactants SDS, APG-06 and Tween-80 had a relatively poor ability to enhance the lignite dust’s wettability, with contact angle reduction rates in the 53%–65% range.

Comment: Here, the contact angle reduction rates of the three surfactants are between 53% and 65%, but the specific concentration is missing. The specific data cannot be directly obtained from the figures. Therefore, this description could be rewritten as "at the CMC of the three surfactants, the proportion of contact angle reduction is...".

8.In the text: 3.3 Effect of compound surfactants on coal dust wettability, first paragraph. Thus, these five surfactants, with a mass fraction of 0.5 wt%, were chosen for 1:1 blending.

Comment: Why choose 0.5wt% surfactant for compounding here, and why not use it directly when the critical micelle concentration has been measured earlier?

9.In the text: 3.3 Effect of compound surfactants on coal dust wettability, second paragraph. The surface tension of CDEA blended with the other four surfactants decreased by 1–2 mN/m.

Comment: This statement is inaccurate. Since there are no specific data in the figure, it is impossible to tell from your description which surfactant has a surface tension reduction of 1 - 2 mN/m.

10.In the text: References

Comment: The spacing between words needs to be adjusted.

Reviewer #2: The manuscript is generally well-written, with a clear structure and logical flow. The abstract provides a concise summary of the study, and the introduction effectively sets the context and justifies the research. The materials and methods section is detailed enough for other researchers to replicate the experiments. The figures and tables are of good quality and help to illustrate the key points. However, the following problems remain.

1.What is the basis for selecting sample particle size? Is the particle size characteristic consistent with the actual dust generated in the mine?

2. Surfactants have the characteristic of being highly efficient in small amounts, and the concentration distribution of surfactants in this study should be several orders of magnitude, such as 10/100/1000 times, in order to have practical significance and scientific validity.

3�“For infrared spectroscopy, we mixed 2 mg of lignite dust with 200 mg of KBr in an additional mortar and ground to below 2μm. We used the KBr tablet method for infrared spectrum detection of the lignite dust.” Is there a corresponding reference for this method?

4.In the section of research status: there are fewer relevant summaries of the latest research status at home and abroad in the article, and it is recommended that additional explanations be given for the current status of domestic and international research in the field of coal dust management. In particular, the research on the wetting of coal dust.

5. In the introduction of the paper, it is mentioned that the innovation of this work lies in the fact that “although extensive research has been conducted on the synergistic effects of surfactant combinations on coal dust wettability, practical engineering applications still limited”. Do you have any engineering case studies for this work?

6. The statement in the proximate analysis section is not very accurate. Under what benchmark were the samples tested? Moisture and fixed carbon are content, while the other two are yield. Also, what is the basis for 'with free water constituting a larger share'?

7.In section 3.3�Why use a monomer solution with a mass fraction of 0.5 for compounding? The author mentioned earlier that high concentrations can form micelles and affect wetting, which seems contradictory to the author's previous description? And the overall concentration of the two solutions has reached 1.0, far exceeding the experimental parameters. According to the author, is it still necessary to conduct previous parameter tests? Isn't it better to directly choose a higher concentration to achieve the goal? Is there a reference for the 1:1 compounding ratio? Have you considered other compounding ratios?

Reviewer #3: 1.The research methodology is sound and appropriate, and the writing is clear and concise. Conclusions or summaries are accurate and supported by content. The article is relevant to members of the educational research community.

2.In the introduction section, it is recommended to enrich the state of the research section by summarising and outlining more cutting-edge applications in the field, thus highlighting the importance and novelty of the research.

3.Is there any latest data in the first paragraph of the introduction? As is well known, it is already 2025, and the statistical data is still stuck in 2023.

4.Why is this particle size used in all experiments for 80 mesh �0.178mm�? What is the basis for proposing? Is there any special significance?

5.The ordinate axis of Figure 3 is not marked with any labels and units, which makes it impossible to judge the data dimension and numerical range. The lack of coordinate information will affect the reader 's interpretation of the data trend. Please modify.

6.The conclusions section is all qualitative, please add quantitative conclusions.

Reviewer #4: This article takes lignite as the research object, combines various methods to obtain its physical and chemical properties and wettability related parameters, and selects 12 different surfactants from three major categories for optimization and compounding. Combining surface tension, contact angle, and settling rate parameters, the optimal compounding solution is finally determined. Overall, the content is relatively rich, but further theoretical analysis is needed. The author is requested to refer to and answer some questions and suggestions raised in this regard.

i. Why are the seven concentrations of 0.01,0.03,0.05,0.07,0.1,0.3,0.5 chosen for the formulation of surfactants, and is there a corresponding reference for such concentrations?

ii. The clarity of Figure 3 is not very good, please further improve the quality of Figure 3.

iii. Some references were found to be inconsistent in the manuscript, please check all reference styles.

iv. Figure 1 is not aesthetically pleasing, please make adjustments to the picture arrangement

v. The serial numbers in figures 1, 5 and 7 are not labelled in the same way, please harmonise them.

6. PLOS authors have the option to publish the peer review history of their article (what does this mean? ). If published, this will include your full peer review and any attached files.

**Do you want your identity to be public for this peer review?** For information about this choice, including consent withdrawal, please see our Privacy Policy .

Reviewer #1: No

Reviewer #2: No

Reviewer #3: No

Reviewer #4: No

---

## [Author Response · Author response to Decision Letter 1]

23 Jun 2025

Response to the editor and Reviewers

Dear editor and reviewers:

I am very much thankful to the editor and reviewers for their deep and thorough review. I have revised my present research paper in the light of their useful suggestions and comments(Tracked changes are used to indicate what was modified). I hope my revision has improved the paper to a level of their satisfaction. Number wise answers to their specific comments/suggestions/queries are as follows.

Editor

Comments 1�Please ensure that your manuscript meets PLOS ONE's style requirements, including those for file naming. The PLOS ONE style templates can be found at

Author Reply: Thanks to your suggestions, we've formatted the entire manuscript according to the template you provided.

Comments 2�In your Methods section, please provide additional information regarding the permits you obtained for the work. Please ensure you have included the full name of the authority that approved the field site access and, if no permits were required, a brief statement explaining why.

Author Reply: The data in the paper were obtained from our experiments and the experimental instruments in Fig. 1 were purchased by the first unit, Kunming University of Science and Technology, and we can use them normally for scientific research purposes.

Comments 3�Thank you for stating the following financial disclosure:

“This research was financially supported by the Teacher category project of Yunnan Provincial Department of Education Scientific Research Fund (grant No. 2023J0158), Yunnan Fundamental Research Projects (grant NO. 202401AU070195), Scientific Research Fund Projects of Yunnan Provincial Department of Education (grant NO. 2024Y133), Teacher category project of Yunnan Provincial Department of Education Scientific Research Fund (grant No. 2023J0157)�Key Research and Development Plan of Yunnan Province�grant No. 202303AA080014�, College Students' Innovative Entrepreneurial Training Plan Program�grant No. 202310674017�.”

Author Reply: These funders provide financial support for experimental research and play a role in the decision to publish the manuscript.

Comments 4�We note that you have indicated that there are restrictions to data sharing for this study. PLOS only allows data to be available upon request if there are legal or ethical restrictions on sharing data publicly. For more information on unacceptable data access restrictions, please see http://journals.plos.org/plosone/s/data-availability#loc-unacceptable-data-access-restrictions.

Author Reply: Thank you for the suggestion that the datasets used and/or analyzed in this study are available upon request from the corresponding author and that we upload the data as supporting information files.

Comments 5�PLOS requires an ORCID iD for the corresponding author in Editorial Manager on papers submitted after December 6th, 2016. Please ensure that you have an ORCID iD and that it is validated in Editorial Manager. To do this, go to ‘Update my Information’ (in the upper left-hand corner of the main menu), and click on the Fetch/Validate link next to the ORCID field. This will take you to the ORCID site and allow you to create a new iD or authenticate a pre-existing iD in Editorial Manager.

Author Reply: Thank you for your suggestion, I have updated my ORCID id in the system!

Reviewer 1

This study proposes a set of solutions to the coal dust pollution problem in Xiaolongtan Town. Specifically, it first analyzes the physical and chemical properties of the coal dust, and then optimally selects the dust suppressants suitable for the local coal dust through experiments. However, there are some issues with the language description in the article, which may cause confusion.

Detailed specific comments

Comments 1� In the text: 1 Introduction, first paragraph. In coal dust suppression. Surfactants are often wetting agents because they reduce surface tension and enhance wettability [10].

Comment: The main content of this study is to solve the problem of coal dust pollution. Therefore, the description here can be written as "reducing the surface tension of water and enhancing the wettability of coal dust."

Author Reply: Thanks to your valuable suggestions, we have revised the statements in the manuscript.

Comments 2�In the text: 2.2 Experimental methods and equipment, second paragraph. We pressed the ground coal dust into thin tablets using an HY-12 tablet press mould (Tianjin Tianguang Optical Instrument Co., Ltd., Tianjin, China) at a pressure of 20 MPa for 2 min.

Comment: When pressing tablets for contact angle measurement, the specific dimensions of the pressed coal tablets should be stated.

Author Reply: The mold used for pressing is the HF-2H mold produced by Tianjin Tianguang Optical Instrument Co., Ltd. of China, and the diameter of the pressed coal piece is 30mm and the height is 5mm.

Comments 3�In the text: 2.2 Experimental methods and equipment, second paragraph. We measured the contact angles between the surfactant solutions or distilled water and the coal sample tablets using a ZJ-6900 optical water droplet contact angle metre (Zhijia Instrument Co., Ltd., Shenzhen, China, as shown in Fig. 1e) with the sessile drop method.

Comment: When measuring the contact angle using the sessile drop method, the choice of which second of data should be expressed.

Author Reply: When using the contact angle measuring instrument for contact angle determination, first we will video record the whole process, choose the contact angle test photo 1s after the drop of the solution for contact angle calculation, each experimental samples measured three times to take the average value.

Comments 4�In the text: 3.1 Influence of physico-chemical properties of coal dust on lignite dust wettability, second paragraph. The total proportion of free and bound water in the sample was 23.76%, with free water constituting a larger share.

Comment: In your figures, there is no visible indication that free water accounts for a relatively large proportion. If this analysis is based on past experience, relevant references should be added.

Author Reply: Thank you for your suggestion, this analysis is based on past experience and we have added relevant references to the manuscript.

Comments 5�In the text: 3.2.1 Surface tension of monomer surfactants, fourth paragraph. Non-ionic surfactants AEO-9 and CDEA can quickly reach their CMC and significantly reduce the solution’s surface tension at CMC concentration, with 54.48% and 55.94% reduction rates, respectively.

Comment: CMC already represents the critical micelle concentration, so "CMC concentration" is a redundant expression.

Author Reply: Thanks to your suggestion, we have corrected this issue in the manuscript.

Comments 6�In the text: 3.2.2 Contact angle of monomer surfactants with lignite, second paragraph. As the concentration of the solutions increases, the contact angle between the solution and the lignite tablets gradually decreases. At room temperature (25°C), the contact angle between distilled water and the lignite tablet was measured to be 87.81°, confirming the poor lignite dust wettability.

Comment: These two sentences lack logical coherence. The first part describes that the contact angle decreases as the solution concentration increases, while the second part mentions that the contact angle of water at room temperature is quite large. There is no connection with the description in the third paragraph about surfactants reducing the contact angle. It would be better to swap the order of these two sentences and combine them with the content in the third paragraph for a more coherent narrative.

Author Reply: Thank you for your suggestion, we do have some problems with our logic here, and we have swapped the two sentences to make the context coherent.

Comments 7�In the text: 3.2.2 Contact angle of monomer surfactants with lignite, fourth paragraph. However, surfactants SDS, APG-06 and Tween-80 had a relatively poor ability to enhance the lignite dust’s wettability, with contact angle reduction rates in the 53%–65% range.

Comment: Here, the contact angle reduction rates of the three surfactants are between 53% and 65%, but the specific concentration is missing. The specific data cannot be directly obtained from the figures. Therefore, this description could be rewritten as "at the CMC of the three surfactants, the proportion of contact angle reduction is...".

Author Reply: Thank you for providing us with important advice, there was a missing condition at the beginning of this process and we have corrected this error.

Comments 8�In the text: 3.3 Effect of compound surfactants on coal dust wettability, first paragraph. Thus, these five surfactants, with a mass fraction of 0.5 wt%, were chosen for 1:1 blending.

Comment: Why choose 0.5wt% surfactant for compounding here, and why not use it directly when the critical micelle concentration has been measured earlier?

Author Reply: After the monomer surfactant solution was compounded in the ratio of 1:1, the mass fraction of individual surfactant solutes was halved to 0.25 wt %, which is higher than the critical micelle concentration of each surfactant analyzed by the monomer experiments, and the synergistic effect of the compounding of the two monomer surfactants after reaching the critical micelle concentration was explored.

Comments 9�In the text: 3.3 Effect of compound surfactants on coal dust wettability, second paragraph. The surface tension of CDEA blended with the other four surfactants decreased by 1–2 mN/m.

Comment: This statement is inaccurate. Since there are no specific data in the figure, it is impossible to tell from your description which surfactant has a surface tension reduction of 1 - 2 mN/m.

Author Reply: Thank you for suggesting that the surfactant CDEA, when compounded with the other four surfactants, decreased the surfactant by 1 to 2 mN/m compared to its monomeric surfactant.

Comments 10�In the text: References

Comment: The spacing between words needs to be adjusted.

Author Reply: Thank you for pointing out the problem for me, I have reformatted the references in the manuscript.

Reviewer 2

The manuscript is generally well-written, with a clear structure and logical flow. The abstract provides a concise summary of the study, and the introduction effectively sets the context and justifies the research. The materials and methods section is detailed enough for other researchers to replicate the experiments. The figures and tables are of good quality and help to illustrate the key points. However, the following problems remain

Comments 1� What is the basis for selecting sample particle size? Is the particle size characteristic consistent with the actual dust generated in the mine?

Author Reply: A stainless steel metal plate measuring 600 mm × 400 mm was positioned 10 m from the working platform to capture dust generated during the operation. The collected dust was analyzed using a 1076 laser particle size analyzer (Furbs Detection Equipment Co., Ltd.). The analysis revealed a large proportion of dust particles within the size range of 0.100 to 0.300 mm. Consequently, lignite dust with a particle size of 0.178 mm, which meets the 80-mesh screening standard, was selected for further study.

Comments 2�Surfactants have the characteristic of being highly efficient in small amounts, and the concentration distribution of surfactants in this study should be several orders of magnitude, such as 10/100/1000 times, in order to have practical significance and scientific validity.

Author Reply: In this study, seven concentrations of monosurfactants (0.01, 0.03, 0.05, 0.07, 0.1, 0.3, and 0.5 wt%) were selected for the following reasons:

1. Concentrations of 0.01, 0.1, 1, and 10 wt% could not effectively prove the efficiency of surfactants at low concentrations. At 0.1 wt%, the surfactants achieved their maximum effectiveness in reducing surface tension.

2. The selected concentrations enabled the analysis of the critical micelle concentration (CMC) for each monosurfactant. Additionally, increasing the concentration from 0.01 to 0.03 wt% significantly enhanced the ability of surfactants to reduce surface tension, indicating that the surfactants are effective even at low concentrations.

Comments 3�“For infrared spectroscopy, we mixed 2 mg of lignite dust with 200 mg of KBr in an additional mortar and ground to below 2μm. We used the KBr tablet method for infrared spectrum detection of the lignite dust.” Is there a corresponding reference for this method?

Author Reply: Yes, the KBr tablet method is frequently used in numerous studies, such as Wang Xiaonan's article titled ‘‘Synergistic effect of surfactant compounding on improving dust suppression in a coal mine in Erdos, China’’ and Meng Xiangxi’s article ‘‘Study on Micro Wetting Mechanism and Compound Optimization of Surfactant on Coal.’’ According to the General Rules for Infrared Spectroscopy Analysis Methods in China (GB/T 6040-2019), the KBr compression method is widely used in Fourier-infrared spectroscopy analysis. This can be attributed to the good infrared transparency of KBr, its chemical stability, ease of preparation, and lack of interference with the absorption peaks in the critical region of the infrared spectrum.

Comments 4�In the section of research status: there are fewer relevant summaries of the latest research status at home and abroad in the article, and it is recommended that additional explanations be given for the current status of domestic and international research in the field of coal dust management. In particular, the research on the wetting of coal dust.

Author Reply Thanks to your suggestion, I have summarized and added to the current state of research in the field content in the introduction.

Comments 5�In the introduction of the paper, it is mentioned that the innovation of this work lies in the fact that “although extensive research has been conducted on the synergistic effects of surfactant combinations on coal dust wettability, practical engineering applications still limited”. Do you have any engineering case studies for this work?

Author Reply: The respiratory dust concentration was measured at Mechanical and Electrical Equipment Co., Ltd. at distances of 8, 12, 16, and 20 m from the working platform. Measurements were conducted during production operations after both a water spray and an AES + AEO composite dust suppressant spray were applied. Five measurements were performed at each point, and the results were averaged to ensure accuracy. The results (Figure 1) revealed a significant decrease in dust concentration at the various test points.

Fig. 1

---

## [Decision Letter · Decision Letter 1]

4 Jul 2025

Synergistic effects of surfactant blends on lignite dust wettability

PONE-D-25-21428R1

Dear Dr. Fang,

We’re pleased to inform you that your manuscript has been judged scientifically suitable for publication and will be formally accepted for publication once it meets all outstanding technical requirements.

Kind regards,

Ugur Ulusoy, Ph.D.

Academic Editor

PLOS ONE

Additional Editor Comments (optional):

Reviewers' comments:

Reviewer's Responses to Questions

**Comments to the Author**

1. If the authors have adequately addressed your comments raised in a previous round of review and you feel that this manuscript is now acceptable for publication, you may indicate that here to bypass the “Comments to the Author” section, enter your conflict of interest statement in the “Confidential to Editor” section, and submit your "Accept" recommendation.

Reviewer #1: All comments have been addressed

Reviewer #2: All comments have been addressed

Reviewer #4: All comments have been addressed

2. Is the manuscript technically sound, and do the data support the conclusions?

Reviewer #1: Yes

Reviewer #2: Yes

Reviewer #4: Yes

3. Has the statistical analysis been performed appropriately and rigorously? 

Reviewer #1: Yes

Reviewer #2: Yes

Reviewer #4: Yes

4. Have the authors made all data underlying the findings in their manuscript fully available?

Reviewer #1: Yes

Reviewer #2: Yes

Reviewer #4: Yes

5. Is the manuscript presented in an intelligible fashion and written in standard English?

Reviewer #1: Yes

Reviewer #2: Yes

Reviewer #4: Yes

6. Review Comments to the Author

Reviewer #1: Dear Author,You have proposed the use of a compound surfactant to address the coal dust pollution issue in Xiaolongtan Town, and have conducted experiments to demonstrate that the compound surfactant can effectively reduce dust levels. We find your experimental design to be outstanding and rigorous, offering valuable insights into this area of study. Furthermore, we are very grateful for your diligent revisions to your paper in response to our suggestions and for addressing our questions. Consequently, we are pleased to recommend your paper for publication.Yours sincerely,

Reviewer #2: Through the author 's revision of the manuscript and the response to the questions raised by the reviewer, the paper proposal was published.

Reviewer #4: The author has made accurate revisions based on the feedback provided, and the key issues raised have been highlighted in the article. We agree to publish it in PLOS One journal

7. PLOS authors have the option to publish the peer review history of their article (what does this mean? ). If published, this will include your full peer review and any attached files.

**Do you want your identity to be public for this peer review?** For information about this choice, including consent withdrawal, please see our Privacy Policy .

Reviewer #1: No

Reviewer #2: No

Reviewer #4: No

---

## [Editor Report · Acceptance letter]

PONE-D-25-21428R1

PLOS ONE

Dear Dr. Fang,

I'm pleased to inform you that your manuscript has been deemed suitable for publication in PLOS ONE. Congratulations! Your manuscript is now being handed over to our production team.

Kind regards,

on behalf of

Prof. Dr. Ugur Ulusoy

Academic Editor

PLOS ONE